# Analysis of the Research Methodology in Spanish Doctoral Theses on Handball. A Systematic Review

**DOI:** 10.3390/ijerph182010579

**Published:** 2021-10-09

**Authors:** Antonio Antúnez, Sergio J. Ibáñez, Sebastian Feu

**Affiliations:** 1Research Group in Optimization of Training and Sports Performance (GOERD), University of Extremadura, Av. de la Universidad, s/n, 10005 Caceres, Spain; sibanez@unex.es (S.J.I.); sfeu@unex.es (S.F.); 2Sport Science Faculty, University of Extremadura, Av. de la Universidad, s/n, 10005 Cáceres, Spain

**Keywords:** handball, review, methodology, doctoral theses, Spain

## Abstract

The objective of this investigation was to analyze scientific production assessed by indexed doctoral theses in the Ordered Spanish Theses (TESEO) database, on the topic of the sport of handball in Spain. Productivity was analyzed on the basis of variables grouped by contextual information, methodologies and procedures. Seventy-two indexed theses from between 1976 and 2021 were analyzed. A progressive increase was identified in scientific production based on these theses during this period. The scientific disciplines that presented the highest number of theses were Sport Sciences (*n* = 33) and Sport Pedagogy (*n* = 19). The main results show that quantitative approaches are the most common (*n* = 49), doctoral theses using descriptive studies based on systematic observation represent the majority (73%), the predominant type of data collection was the use of cross-sectional studies (70.8%) versus longitudinal studies (26.4%), and the most used sampling method was that of convenience (*n* = 65). The results make it possible to ascertain the reality of this research topic, the methodological positioning and research tendencies, and to draw the basic lines for development.

## 1. Introduction

At present, the need to transmit scientific knowledge in our society has gained significant importance in different environments (universities, congresses, symposia, etc.), with different academic forms (undergraduate dissertations, master’s theses, and doctoral dissertations), or for dissemination (articles, manuals, and books). The same aspects also exist in sport science research [1].

In Spain, there are institutions that sponsor specialist research groups in different scientific disciplines, develop courses and seminars, and establish programs for the training of young researchers. These institutions that collaborate with research are public and can be national through the Spanish Ministry of Universities by the Order of 6 November 2020, regional, such as those of the Ministry of Economy and Finance of the Government of Extremadura through Decree 56/2021 of 4 June, or through each university’s own grants, also from the Spanish Ministry of Universities through Royal Decree 289/2021 of 20 April. All of them can be accessed through competitive public calls for applications. It is undoubtedly in the institutional context of the university par excellence that the generation of knowledge and the training of individuals at the different academic levels of the degree, graduate studies, master’s degree and doctorate, are encouraged. The doctoral training process culminates with the defense of a doctoral thesis, which implies completing an original scientific project with the aim of showing the scientific community that the doctoral student is capacitated to develop quality research. This is the highest academic level.

One of the indicators that makes it possible to assess a scientific field in general is the production of doctoral theses [2]. It is also one of the indicators of quality for the establishment of rankings of researchers, universities, departments, etc. [3,4]. The completion of a doctoral thesis as well as being a requisite for obtaining the highest academic level, contributes important scientific findings [5]. The analysis of doctoral theses makes it possible to identify the reality of a scientific discipline, its evolution, methodological positioning and research tendencies [1]. Researchers in the different scientific fields develop studies with the aim of updating the progress made in research topics with reviews derived from other studies. There are three formats for these reviews [6]: (i) narrative reviews (a subjective theoretical review of primary studies on a research topic, without empiric contributions from the researcher); (ii) systematic reviews (reviews of primary studies, with a systematic development of the data collection process, where statistical procedures are not used to summarize the studies; (iii) quantitative systematic reviews or meta-analyses (overview of the primary studies with quantitative methodology that contains both a systematic development of the data collection process and the use of statistical methods to summarize the studies).

To answer a concrete research question, systematic reviews collect the main evidence that comply with the previously specified eligibility criteria [7]; thus, they limit the studies to different scientific disciplines and research topics [8], as well as the reality and context of each country [9,10,11], or to a concrete sport discipline [12,13]. It is imperative to delimit the studies in a systematic review to precisely understand the current situation regarding a research topic in the local and international context, generating the development of comparative studies among countries [1]. In this regard, the study by Ibáñez and Feu [14], which presents a comparative review of the scientific literature on the sport of basketball between Spain and Portugal, should be highlighted. Moreover, there are studies that develop reviews on research in the field of game analysis [15,16]. Similarly, other analyses focus on systematic reviews on a sport discipline such as volleyball [17], soccer [18,19,20], or basketball [21].

In handball, systematic reviews have been identified that specifically address some game action, such as match analysis [22]; the analysis of throwing speed [23]; injury profiles in handball players [24]; or physical and psychological performance factors [25].

The majority of systematic reviews in sport sciences are focused on analyses of topics of scientific interest disseminated in research articles with quality indicators. The searches are based on indexed articles in reference databases (WOS, Scopus, and Pubmed). It cannot be said that there are a large number of systematic reviews of doctoral theses in sport science topics. Yaman and Atay [11] analyzed the development of the production of doctoral theses on physical education and sport. In Spain, we can find reviews on the incidence of women completing and directing doctoral theses in sport sciences [10], with reference to the history of physical education and sport [26], or about the state of research in basketball [12] or the topic of the sport coach [1]. Publications can also be found related to reviews on the production of doctoral theses in other areas of knowledge such as Education Theory [27], Psychology [5], or Psychiatry [28]. The scarcity of systematic reviews in doctoral theses is due to the difficulties that researchers have to access this type of primary documents. However, the scientific knowledge produced ends up being transmitted to society because many Ph.D.s take advantage of the results of their theses to disseminate them partially or in toto in scientific journals.

Spain, as an active member of the processes leading to the creation and development of a European knowledge area, has been incorporating the legislative reforms that have enabled it to consolidate a range of courses in accordance with the principles of the EHEA. Thus, official doctoral studies are regulated by Royal Decree 99/2011 of January 28. A doctoral thesis is an original research work related with one of the fields of knowledge (scientific, humanist, biomedical, social and technical) that the doctoral student will defend in front of a panel in public once they have completed their training in a doctoral program.

In sport sciences, the bibliometric study of doctoral theses is of great interest for analyzing and observing the state of the discipline on which researchers are working, especially if it is desired to consolidate knowledge as it shows a specific development, as well as to include in the body of knowledge, if desired, new constructs and theories that help to better understand the different ambits of the different disciplines [29,30].

In this respect, it must be pointed out that the study of handball, in particular, has seen significant growth in the last two decades. Not only has there been a marked increase in the number of doctoral theses in Spain that have this discipline as their object of study, but also in the organized academic events (congresses, seminars, workshops, forums, etc.), and in the books and research articles published with special mention of the specific scientific journal E-Balonmano com, indexed in the Emerging Sources Citation Index and recently in SCOPUS.

Faced with the scarcity of research that analyzes the reality of a scientific discipline or research topic in a systematic review of the production of doctoral theses, it is imperative to increase this type of study as in other sport disciplines [1]. Thus, the objective of the present study was to analyze the production of doctoral theses in Spain that are centered on handball as their main object of study, bearing in mind the research classification, procedures and relations. The study has therefore been carried out using the analysis of documents [31] included in the Ordered Spanish Theses Database TESEO.

## 2. Methodology

### 2.1. Design

The compilation of the reports on progress in research on handball using doctoral theses presented in Spain uses a theoretical research design [6]. It is a systematic review with a methodological development to obtain the data (compilation of studies, specification of variables, cataloging, analysis, etc.). The research followed the Preferred Reporting Items for Systematic reviews and Meta-Analyses (PRISMA) [7] guidelines determined as: (i) definition of the objectives with explicit and reproducible methodology; (ii) systematic search for evidence following eligibility criteria; (iii) assessment of the validity of the findings; and (iv) systematic presentation and synthesis of the characteristics and findings of the included studies.

### 2.2. Sample

Data were collected from the TESEO database (Tesis Españolas Ordenadas, https://www.educacion.es/teseo, accessed on 31 January 2021), which is generated by the Spanish Ministry of Education, Culture and Sport, and indexes all the doctoral theses completed and defended in public and private universities since 1976. It is compulsory for newly created Ph.D.s to have the general data on their theses indexed, for the general knowledge of the whole scientific community. This database makes it possible to search using five fields, title, author, identity card number, university and academic year.

### 2.3. Procedure

The systematic review of scientific literature demands the use of precise procedures and tools that make it possible to present evident scientific contributions [32]. The methodology of this study followed the PRISMA guidelines for systematic reviews [7] (Figure 1), divided into four phases: (I) identification; (II) screening; (III) eligibility; (IV) included.

During phase I, a total of 60 doctoral theses were initially identified in the database including the term “handball” and limiting the search to the field of “title”, making it possible to identify the object of this review generically. It was found that the number increased to 121 if the abstract was included in the search, and 123 if title and abstract were included. Due to the possibility of defending doctoral theses in Spain in different languages, it was decided to include the term handball in the globally recognized languages of English, German, Portuguese and French, and the languages recognized in the Spanish constitution, Galician, Basque and Catalan, identifying 13 doctoral theses and bringing the number to 135. Two theses were found in Google Scholar that were not indexed in the TESEO database such that the total increased to 137. Doctoral theses are not indexed immediately, such that the database may not be completely up to date. For this reason, the search was made at two time points, on 31 January 2021 and on 30 June 2021, when the search was terminated.

In Phase II, the titles, abstracts and UNESCO descriptors included in each thesis found were read, leading to their screening to eliminate those that were not on the topic of handball, selecting only those in which the authors included handball in their research, and discriminating between the theses that exclusively studied handball and those that included it with other sports. In certain cases, the term “handball” appeared in the abstract as secondary to the study or simply in a reference; thus, precise inclusion and exclusion criteria were determined.

Four inclusion criteria were defined: (i) doctoral theses included in the TESEO database with the term “handball” in the title in globally recognized languages (Spanish, English; German, French and Portuguese) and in the languages recognized by the Spanish Constitution; (ii) handball as the main object of the study, handball as the only sample as the object of study, handball as a sample among other sports; (iii) accessibility to the document in the TESEO database or to the complete thesis if necessary; (iv) produced in the period between 1976 and 2021.

The two exclusion criteria were: (i) the sport of handball was not the object of study and/or the term handball was not included in the thesis title; (ii) failure to access the document in the TESEO database or the complete thesis if necessary.

After applying the inclusion criteria and coding all the doctoral theses, the sample was reduced to 72 documents in which the term handball was included in the thesis title.

In Phase III, the documents retrieved from the TESEO database were analyzed. On occasions, the summary in the database did not contain sufficient information to identify all the variables of the study. In this case, the complete document of the doctoral thesis was sought in Google Scholar and the archive of doctoral theses in the university where it was publicly defended. Once the complete theses had been located, none had to be excluded in this phase.

Finally, in Phase IV, the identified theses were analyzed, according to 13 previously defined variables.

### 2.4. Variables

Thirteen variables were established divided into contextual, methodological and procedural (Figure 2).

Seven contextual variables were coded for each doctoral thesis: (a) author’s name; (b) author’s sex; (c) year of defense; (d) title of thesis; (e) name of director/s; (f) sex of director/; (g) university where defended.

Three variables were used to code the methodological variables: (h) classification of the scientific disciplines defined by Borms [33], in the catalogue in the Directory of Sport Science (5th edition) which are: (i) Adapted Physical Activity, (ii) Biomechanics of Sport, (iii) Coaching Sciences, (iv) Kinanthropometry, (v) Motor behavior, Motor development, Motor control and Motor learning, (vi) Philosophy of sport, (vii) Sociology of sport, (viii) Sport and Exercise Physiology, (ix) Sport and Exercise Psychology, (x) Sport and Leisure Facilities, (xi) Sport History, (xii) Sport Information, (xiii) Sport Law, (xiv) Sport Management, (xv) Sports Medicine, (xvi), Sport Pedagogy; (j) research approaches suggested by Ibáñez et al. [1]: (i) qualitative, (ii) quantitative, (iii) mixed, (iv) intervention; and (k) research methods proposed by Montero and León [34]: theoretical study, (i) classic, (ii) meta- analysis; empirical study with quantitative methodology, (iii) descriptive study using systematic observation, (iv) descriptive study of populations via research by surveys, (v) experiments, (vi) quasi-experiments, (vii) ex post facto studies, (viii) experiments with a single subject, (ix) instrumental studies; empirical qualitative studies: (x) ethnography, (xi) case studies, and (xii) research-action. This variable was coded as having multiple answers as different research methods can be applied.

Finally, three variables were used to code the research procedures: (l) a method of data gathering based on the proposals of Nelson and Silverman [35], Hernández et al. [36] and Polit and Hungler [37]. This variable was coded as having multiple answers, as a thesis can be carried out using different data collection methods. The categories into which this variable was divided were: (i) questionnaire/scale, (ii) interview, (iii) systematic observation, (iv) qualitative observation/field notes, (v) documents, (vi) tests, (vii) discussion groups, (viii) self-report and others identified by the authors, (ix) training program, (x) teaching units, (xi) and electronic devices; (m) for the type of sampling the categorization by Cubo was used [38]: probability: (i) simple random, (ii) stratified random, (iii) systematic random, (iv) random cluster, (v) multistage; and non-probability: (vi) convenience, (vii) intentional, (viii) quasi-probability, (ix) hard to reach populations, (x) by quotas; and (n) depending on the type of information or data that are obtained as proposed by Cubo [39]: (i) cross-sectional, (ii) longitudinal l, or (iii) both if the two possibilities are used in the same study.

A formula for coding was created for each doctoral thesis and stored by date of defense/reading.

### 2.5. Reliability

Three researchers who knew the document analysis system performed the coding. The reliability of the coding was tested with the free-marginal multirater kappa [40]. Ten per cent of the doctoral theses were analyzed to calculate the agreement among the coders. The score obtained for each variable can be considered “almost perfect” [41], with an average value of 0.95. Four variables were identified with 100% agreement (author’s sex, type of sampling, research methods, type of information) and three variables between 83.33% and 91.67% agreement (research procedure [35,36,37], classification of scientific disciplines following Borms [33] and research approach [1]).

### 2.6. Data Analysis

A descriptive analysis was made of all the variables recorded in the research (frequency and percentage). An analysis was also made of the multiple answers in the following variables: director/s, sex of director/s; research method; data compilation method. These variables in the contingency tables may surpass 100%, as there are variables with multiple answers. The analysis was completed with contingency tables to identify the relations among the study variables.

## 3. Results

Seventy-two doctoral theses were identified for this systematic review that mainly focused on handball (Annex 1). Eighty-four point seven percent of the authors were men (*n* = 61), and 15.3% women (*n* = 11). This difference is maintained regarding the directors as 82.9% were men and 17.1% were women.

The first doctoral thesis that fulfilled the inclusion criteria coincided with the first one devoted to handball in 1991. The production of doctoral theses on handball shows growing interest in the Spanish university context, as there has been a linear progression in production (R^2^ = 0.2899), increasing by 28.99% in the period analyzed.

Doctoral theses on handball have only been defended in 39% (*n* = 34) of Spanish universities. Those that have developed the most are the University of Granada (UGR) and the University of Coruña (UdC) with 9.7% each (*n* = 7), followed by the University of Lérida (UdL) with 7% (*n* = 5) and the Universities of Barcelona (UB), Vigo (UVIGO) and Valencia (UV), with 5.5% each (*n* = 4).

### 3.1. Methodological Variables

According to the classification of scientific disciplines by Borms [33] as the criterion for classifying doctoral theses on the topic of handball, it can be seen that Coaching Sciences (45.8%) and Sport Pedagogy (26.4%) are the most commonly studied disciplines (Figure 3), dominating the rest.

There are a large number of doctoral theses that use a quantitative research approach (70.84%; *n* = 51) regarding the topic of handball. This number increases if it is considered that 11.1% (*n* = 8) use a mixed research approach (qualitative and quantitative). The rest of the investigations employed a qualitative approach (5.6%; *n* = 4) or included an intervention (12.5%; *n* = 9).

Following the classification proposed by Montero and León [34], the most commonly used research methods are descriptive with systematic observation (48.1%). They are followed by four less usual methods: quasi-experimental studies (12.3%), descriptive studies of populations via surveys (11.1%); instrumental studies (9.9%), case studies (7.4%) and ex post facto studies (6.2%) (Figure 4).

Figure 5 shows the relations in the research methodological variables among the classification of scientific disciplines, the research approaches and the research methods. The main finding is that quantitative approaches are the most common (*n* = 49), doctoral theses that use descriptive studies based on systematic observation represent the majority (73%), and Sport Sciences is the discipline that includes the most studies on handball (55.1%).

### 3.2. Procedural Research Variables

The doctoral theses analyzed mainly used convenience sampling (91.7%; *n* = 66), one case used simple random sampling and another intentional sampling with identical percentages (1.4%; *n* = 1). In the remaining cases, the type of sampling was not specified (5%; *n* = 4).

The doctoral students used different instruments to obtain their data (Figure 6), and sometimes several instruments were used in the same investigation. The most commonly used instrument was systematic observation (27.4%), followed by questionnaires/scales and tests (15.9%), electronic devices (13.3%), interviews (10.6%) and training programs (8%), leaving documents (4.4%,), field notes and discussion groups (1.8%), and self-report (0.9%).

The predominant type of data collection was using cross-sectional studies (70.8%; *n* = 51) compared to longitudinal studies (26.4%). Only one study used both types of data.

Figure 7 shows the relations in the research procedures among methods for data collection, type of sampling and type of information or data. There is a strong predominance of convenience sampling (*n* = 65) over the rest, with cross-sectional theses being the most frequent (*n* = 51). In these studies, systematic observation is the most commonly used method for collecting data (*n* = 18), followed by questionnaires and tests (*n* = 14 in each case) and electronic devices (*n* = 11). Theses with longitudinal data collection are also evident although fewer in number (*n* = 19), with systematic observation being the data gathering method most commonly used (*n* = 12).

### 3.3. Interaction of the Methodological Variables with the Procedural Variables

Table 1 shows the results relating the methodological and procedural variables bearing in mind the search methods, data collection methods and type of sampling. It should be mentioned that there is a clear predominance of convenience sampling in practically all the search methods, except the classic one, the most frequent studies being descriptive using systematic observation. A small percentage used simple random and intentional sampling.

Table 2 presents the interaction of data between methodologies and research procedures bearing in mind the variables of type of information, the Borms classification [33] and the data collection instruments. Regarding cross-sectional information, studies in Coaching Sciences are the most prolific (64%, *n* = 23), followed by studies in Sport Pedagogy (26%, *n* = 13). Only a few studies correspond to Sport Psychology (8%, *n* = 4), Sports Medicine and Kinanthropometry (6%, *n* = 3 in both cases), Sport Biomechanics (4%, *n* = 2) and Motor Behavior (2%, *n* = 1).

Longitudinal studies are found exclusively in three types of scientific disciplines, with again Coaching Sciences the most prolific (55%, *n* = 10), followed by Sport Pedagogy (33.3%, *n* = 6) and finally Sports Medicine (11%, *n* = 2). There is only one thesis that used both types of information, and it is in the scientific discipline of Sport Psychology.

## 4. Discussion

The purpose of this research was to analyze the production of doctoral theses in Spain from 1976 to 2021, recorded in the TESEO database, that focus on the study topic of handball, tendencies, methodological positioning according to the research designs and procedures used, and their evolution.

Agreement among coders was almost perfect [41], showing that that the definition of the variables was adequate. Similar variables were used in the studies by Ibáñez et al. [1] and Gamonales et al. [42]. Agreement in this investigation obtained similar values to those of the studies by Gamonales et al. [42], Gómez-Carmona et al. [43] and Reina et al. [44], which were higher than 0.90 in the different Kappa coefficients. Thus, the information used for this review is reliable.

Doctoral theses on handball are relatively recent in Spain. To date, 72 theses have been identified since the first thesis defended in 1991 by Gil, Ph.D. in the University of Valencia, the title of which included the term “handball”. In spite of being a relatively new research topic, it has seen a progressive increase over the last thirty years reaching a volume of scientific production in doctoral theses that is still well below that of other more general research themes in the field of Sport Sciences, such as the prevalence of women in the completion and direction of doctoral theses in Sport Sciences (333 theses) [10], referring to the History of Physical Education and Sport (88 theses) [26], but similar to the situation of other sport disciplines such as basketball (75 theses) [12], and higher than other more specific themes such as the sport coach (60) [1]. Considering the presence and relevance of handball in the ambit of Spanish sport, it can be stated that from a scientific perspective the evolution in research via doctoral theses on handball is becoming established compared to the other research topics mentioned. This is due to the increase in sport science faculties, with specific subjects on handball in the Physical Activity and Sport Sciences degree, and thus specialized staff in handball in the faculties, and research master’s degrees and doctoral programs, which have increased the number of Ph.D.s in the context of this sport. Research in sport sciences has stopped being dependent on other scientific disciplines, and at present, it is much easier to research a sport discipline from any viewpoint. Some of the Ph.D.s who completed their doctoral theses on handball have in turn become the directors of other doctoral students, thus consolidating their own lines of research.

Handball is studied mainly from two scientific disciplines [33]. The main one is Coaching Science, in which the research lines are directed toward the analysis of sport performance relating this sport discipline with an analysis of the coach’s behavior, technique, performance indicators or the game itself. Although less predominant, it is also featured in Sport Pedagogy, with the studies based on this discipline investigating coaching, teaching and training, including coaches’ training. These investigations deal both with the ways in which players learn and the knowledge and pedagogical abilities that the coaches need to teach more effectively [45]. The predominance of theses that situate their research in these two scientific disciplines coincide with the studies on doctoral theses on coaches [1] and basketball [12]. They are therefore investigations applied to the study of sports modalities.

Regarding research methods, the data analyzed show the predominance of descriptive studies with systematic observation using observational instruments as the most common method, due to its capacity to provide strong data on the development of the activity as occurs in other sport disciplines [46]. Similarly, the research approach most frequently used by the researchers is quantitative, far above qualitative research, and especially, interventions. Mixed approaches (qualitative/quantitative) are used more frequently than these last two as a consequence of the need to tackle the complexity of the research problems considered in handball with the interdisciplinary research processes that require such approaches. Research using mixed methods is currently a new tendency in scientific methodology that represents the integration of qualitative and quantitative research, which can provide more complete knowledge in sport sciences [47]. This information has to be interpreted from two viewpoints. First, historically, researchers who use a quantitative method are considered to be better [48], particularly researchers into sport value more the tradition of quantitative research [47], and moreover, students in doctoral programs are hardly trained in other research methods, as most of them, or even the training manuals, are mainly quantitative, with great importance given to statistics and their application in training courses [48]. This leads to the accumulation of more descriptive knowledge, a more basic level of research. Qualitative research will make possible better knowledge of complex experiments in the sports field [49] but this is not the tendency that has been observed. Research methods used by doctoral students reflect that the research on this sport discipline is still in its infancy, often involving cases of descriptive research, and should evolve to studies that make it possible to diagnose, predict and prescribe, starting from the results obtained, that is, develop from knowledge of the past to being able to anticipate the future.

The research procedure most used in handball has been descriptive studies, with systematic observation using surveys with questionnaires and mainly convenience sampling. Systematic observation has been one of the most commonly used methods in research on sport training and competition and the coach’s intervention [50]. Instruments generated ad hoc to analyze the coach’s intervention, players’ participation and the development of the game, have made it possible to describe their behavior, enhancing the description of the sport context [1]. These are instruments that are tedious to generate and validate, but easy to use for the researcher [51], being mainly used during training and competition. These data are similar to those identified when analyzing Spanish doctoral theses on basketball, with a predominant use of questionnaires and systematic observation [12]. This is a tendency in research topics in the process of development [1]. Using descriptive studies hinders an in-depth understanding of the coach’s intervention and the participation of the players in the game. Both players and coaches have little knowledge of their behavior, and there is an epistemological gap between knowledge and practice [52], again making it necessary to study one’s own actions. The analysis of the demands of training and the competition, as well as the coach’s intervention, require instruments that are applied ecologically to obtain information. This is one of the interests manifested by the researchers, which is why the use of ad hoc instruments using an observational methodology predominate.

The specific procedures for collecting data on players’ performance are varied. There are tests that are characterized by being of a general nature or specific to the respective sport, presenting results with greater ecological validity and reliability, as used in other studies on team sports such as basketball [53]. Conversely, data collection using a great variety of electronic devices makes it possible to monitor training and competition providing objective and powerful data that, with the advances made in the technology, are progressively increasing and becoming more reliable. Pino-Ortega et al. [54] stated that the technology used is attractive for monitoring performance indicators in team sports given its accuracy in collecting the data; although it is true, access is limited due to high cost, and thus it is difficult to obtain the devices. Finally, training programs that provide data on differences in learning tasks or performance, are mostly designed with specific objectives, the validation of which will help researchers to compare their effects with other studies [55]. The objectives of the investigations condition the instruments used to obtain information. These three groups of instruments allow researchers to tackle the issues that are considered in the investigation with great accuracy.

In the majority of studies, the representative population is obtained with convenience sampling choosing available subjects [38], because of the ease with which researchers can have access to, and monitor, the studied population. It has been shown that, with few exceptions, samples are not random in this field of study [56]. Moreover, there is no indication of uniformity of criteria when specifying the volume of representative data to guarantee their validity. All of this complicates generalizing the results, making it increasingly important to incorporate the sampling error for the study population and the effect size in the statistical analyses [57]. Researchers carry out their investigations on populations that are easier to monitor to guarantee the success of their study. The evolution of research in this sport discipline should occur when researchers introduce samples that make it possible to generalize the results.

Cross-sectional studies are the most frequently used to collect data in this context. Compared with longitudinal studies, they are cheaper, making it possible to work with more individuals, produce immediate results, obtain cooperation from the study subjects more easily, and are less affected by the measuring effect [58]. For these reasons, cross-sectional studies are better adapted to handball research, contributing to a simplicity bias if the data provided in the remaining variables are considered. Having reached this level in research on handball, it would be interesting to carry out longitudinal studies to provide information on the evolution of different aspects related to this sport.

## 5. Conclusions

The results of the analysis of the doctoral theses indexed on the TESEO database have made it possible to characterize the present state of research on this sport discipline, research tendencies, and the methodological positioning through the research designs and procedures used. This analysis makes it possible to appreciate the evolution of research on handball and advance in its development.

Research on handball, as represented in doctoral theses, is based on the study of the past, intent on discovering what happened and why the analyzed events occurred, and should evolve toward the future, investigating in order to predict and try to discover what can happen.

Maturity in research on handball in Spain will be reached when there is evolution from descriptive and correlational studies toward a significant increase in intervention and quasi-experimental studies. Thus, it is necessary to evolve from description to the study of players’ behaviors and thoughts in game actions and the coach’s intervention with more qualitative and mixed designs.

An increase in longitudinal designs will make possible a repeated and ordered observation of coaches and players, and their contribution to the development of the game will provide information to identify the processes and causes that are produced in the evolution of handball in their multiple possibilities.

Regarding future perspectives, it is considered that the adaptation of doctoral programs in Spain, extending and guiding the training of doctoral students toward more complex and reflective research methodologies will favor the completion of studies that provide more complete information for sport disciplines in general and handball in particular.

It is considered important to increase predoctoral grants and aid for research on the part of the competent Spanish authorities to obtain technological equipment that will foment a great improvement in data collection at the qualitative and quantitative level in the study of sport disciplines.

This first article reviewing the state of the art on research through doctoral theses on handball in Spain should have continuity over time by updating this scientific production periodically (5 or 10 years). Likewise, comparative review research should be carried out between different sports modalities, as well as research in doctoral theses on the sport of handball between different countries.

Finally, we have found the following limitations to this research. It is possible that our study is incomplete at the closing date of data collection as there is a delay between the defense of the doctoral thesis and its indexing in the TESEO database. Conversely, the selection of the doctoral theses was carried out exclusively using the search criterion of the word handball in the title of the doctoral thesis. It is possible that there are other doctoral theses on handball in which the authors did not decide to include this word in the title.

## Figures and Tables

**Figure 1 ijerph-18-10579-f001:**
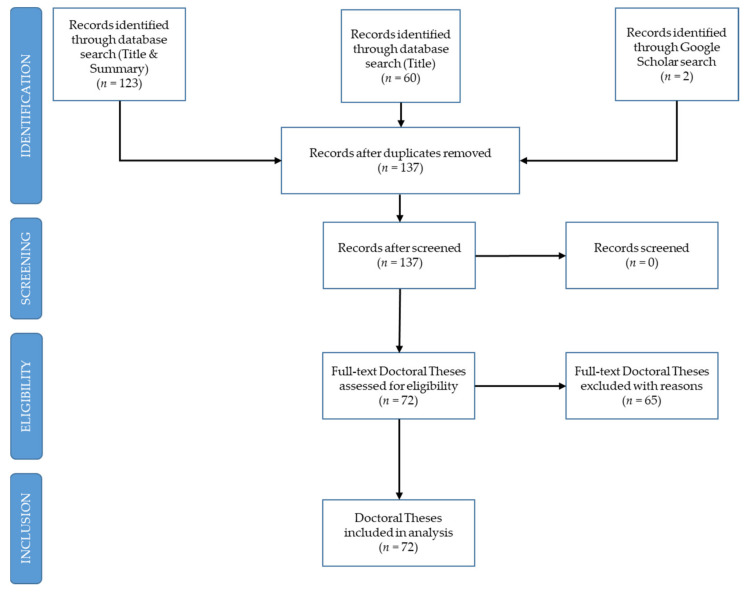
Flow diagram based on the PRISMA guidelines [7] of the methodology used in the search for doctoral theses on handball.

**Figure 2 ijerph-18-10579-f002:**
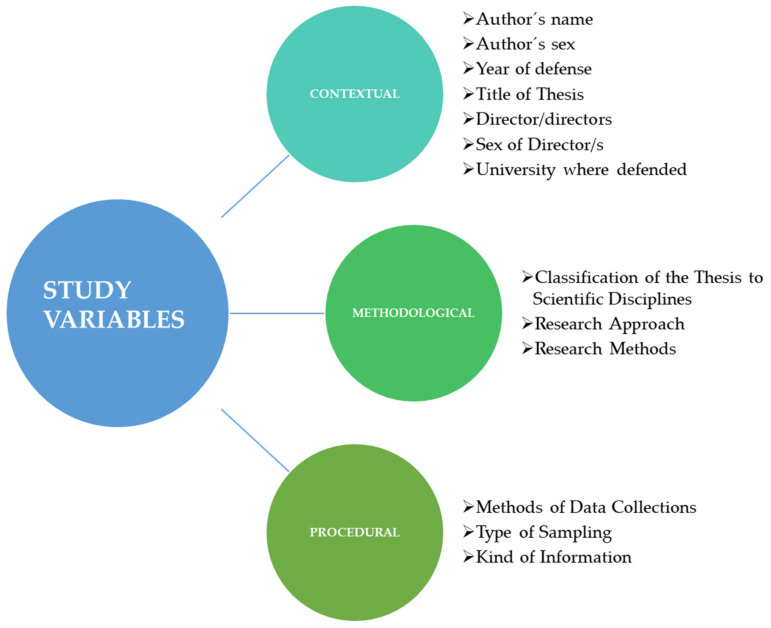
Diagram of the grouping of the study variables.

**Figure 3 ijerph-18-10579-f003:**
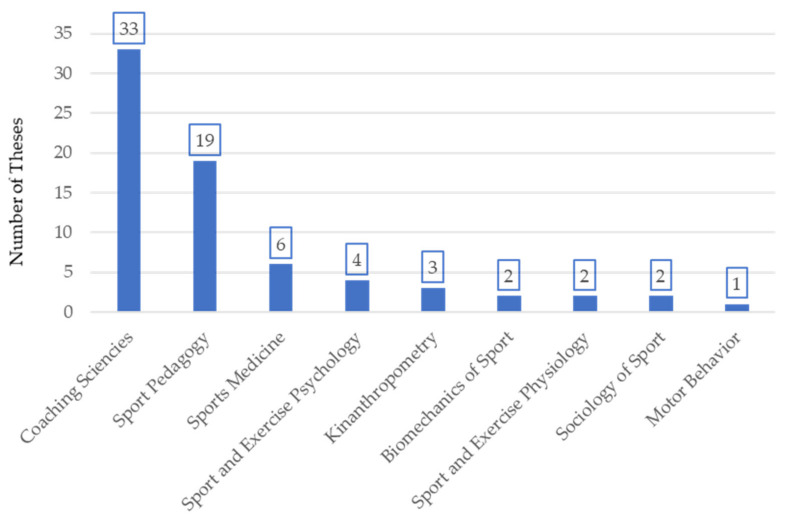
Distribution of doctoral theses on handball by scientific discipline.

**Figure 4 ijerph-18-10579-f004:**
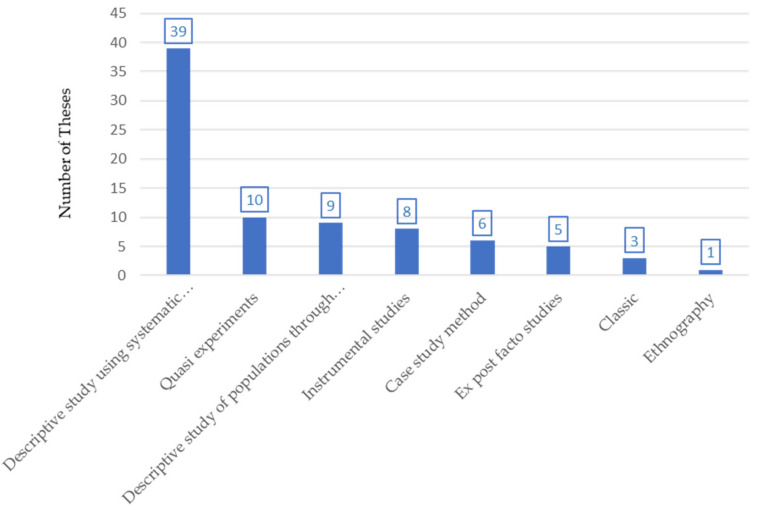
Distribution of the research methods used in the doctoral theses on handball.

**Figure 5 ijerph-18-10579-f005:**
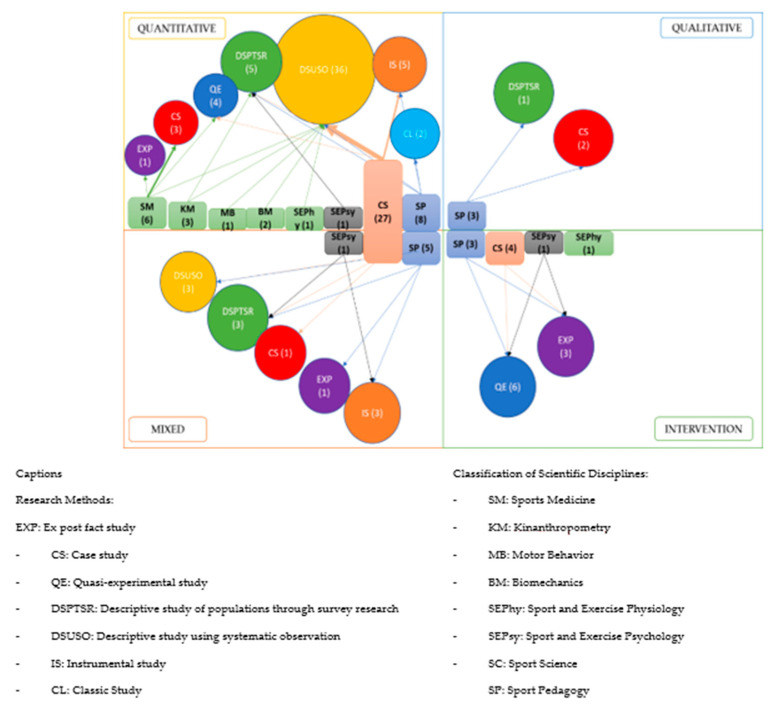
Classification of research in doctoral theses on handball.

**Figure 6 ijerph-18-10579-f006:**
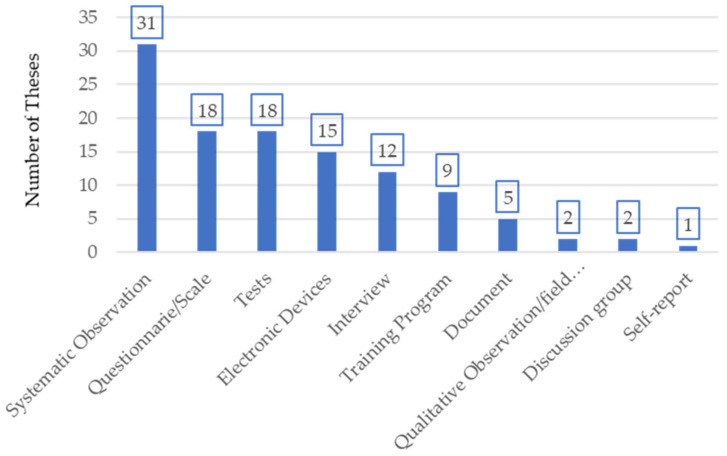
Distribution of the instruments used in doctoral theses on handball.

**Figure 7 ijerph-18-10579-f007:**
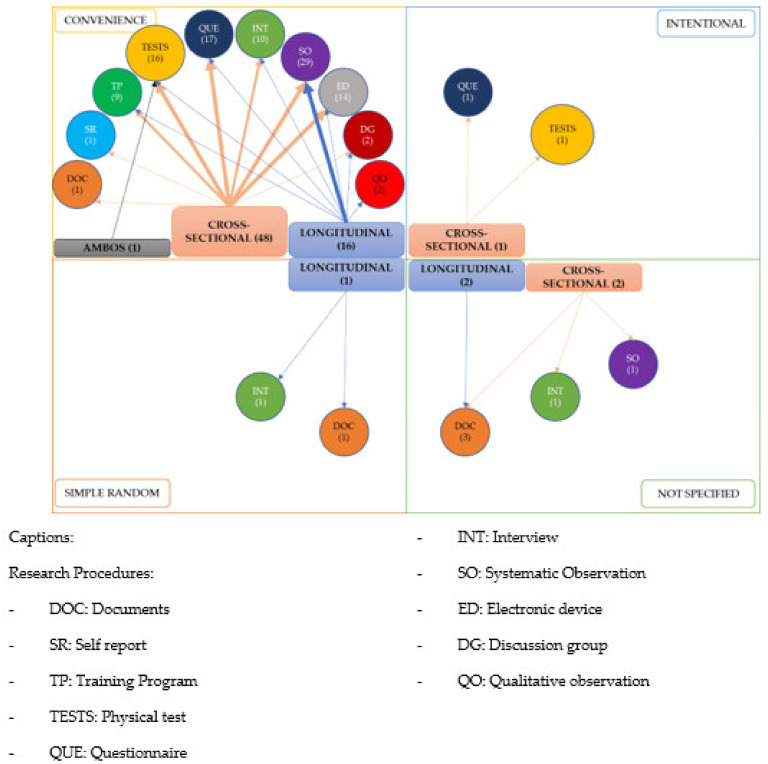
Cross-matching of research procedures.

**Table 1 ijerph-18-10579-t001:** Interaction of data between classification and research procedures relating the variables of search methods, data collection methods and type of sampling.

Research Methods	Type of Sampling		Data Collection Instruments									
			Questionnaire/Scale	Interview	Systematic Observation	Qualitative Observation/Field Notes	Document	Test	Training Program	Electronic Devices	Discussion Group	Self-Report
Classic	Not specified	*n*		1			3					
		%		33.3%			100.0%					
Descriptive study using systematic observation	Convenience	*n*	5	4	26	1		5		8	1	
		%	12.8%	10.3%	66.7%	2.6%		12.8%		20.5%	2.6%	
	Intentional	*n*	1					1				
		%	2.6%					2.6%				
	Not specified	*n*			1							
		%			2.6%							
Descriptive study of populations through survey research	Convenience	*n*	6	3	2		1	1				1
		%	66.7%	33.3%	22.2%		11.1%	11.1%				11.1%
	Intentional	*n*	1					1				
		%	11.1%					11.1%				
Quasi experiments	Convenience	*n*	2		1			4	6	2		
		%	20.0%		10.0%			40.0%	60.0%	20.0%		
Ex post facto studies	Convenience	*n*		1	1			3	2	2		
		%		20.0%	20.0%			60.0%	40.0%	40.0%		
Instrumental studies	Convenience	*n*	4	2	3	2				1	1	1
		%	50.0%	25.0%	37.5%	25.0%				12.5%	12.5%	12.5%
	Not specified	*n*			1							
		%			12.5%							
Ethnography	Simple random	*n*		1			1					
		%		100.0%			100.0%					
Case study method	Convenience	*n*	2	3	2			4	1	3		
		%	33.3%	50.0%	33.3%			66.7%	16.7%	50.0%		

**Table 2 ijerph-18-10579-t002:** Interaction of the data between classification and research procedures relating the variables of type of information, the Borms classification [33] and the data collection instruments.

Type of Information	Borms Classification		Data Collection Instruments									
			Questionnaire/Scale	Interview	Systematic Observation	Qualitative Observation/Field Notes	Document	Test	Training Program	Electronic Devices	Discussion Group	Self-Report
Cross-sectional	Biomechanics of Sport	*n*			1					2		
		%			2.0%					4.0%		
	Coaching Sciences	*n*	3	3	12			5	3	5		
		%	6.0%	6.0%	24.0%			10.0%	6.0%	10.0%		
	Motor Behavior	*n*			1					1		
		%			2.0%					2.0%		
	Sport and Exercise Physiology	*n*						1	1			
		%						2.0%	2.0%			
	Kinanthropometry	*n*	2					3				
		%	4.0%					6.0%				
	Sports Medicine	*n*						2	2	2		
		%						4.0%	4.0%	4.0%		
	Sport pedagogy	*n*	6	4	4		1	2	1	1	1	
		%	12.0%	8.0%	8.0%		2.0%	4.0%	2.0%	2.0%	2.0%	
	Sport and Exercise Psychology	*n*	3	1				1	1			1
		%	6.0%	2.0%				2.0%	2.0%			2.0%
	Total Cross sectional	*n*	14	8	18		1	14	8	11	1	1
		%	28.0%	16.0%	36.0%		2.0%	28.0%	16.0%	22.0%	2.0%	2.0%
Longitudinal	Coaching Sciences	*n*	1	2	9				1	1		
		%	5.6%	11.1%	50.0%				5.6%	5.6%		
	Sports Medicine	*n*	1		1			1		1		
		%	5.6%		5.6%			5.6%		5.6%		
	Sport pedagogy	*n*	2		2	2	2	1		1	1	
		%	11.1%		11.1%	11.1%	11.1%	5.6%		5.6%	5.6%	
	Total Longitudinal	*n*	4	2	12	2	2	2	1	3	1	
		%	22.2%	11.1%	66.7%	11.1%	11.1%	11.1%	5.6%	16.7%	5.6%	
Both	Sport and Exercise Physiology	*n*						1				
		%						100.0%				
	Total both	*n*						1				
		%						100.0%				

## Data Availability

Not applicable.

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
