# Peer review of "Analysis of the Research Methodology in Spanish Doctoral Theses on Handball. A Systematic Review"

_ijerph, 2021, doi:10.3390/ijerph182010579_

Round 1
Reviewer 1 Report
The abstract is well written, but it could be improved if the authors included some of the results obtained, to create curiosity about the research to future readers.
[line 52-54] The authors in the article state: “There are institutions that sponsor specialist research groups in different scientific disciplines, develop courses and seminars, and establish programs for the training of young researchers.” - but what are these institutions? Where did the authors get access to this information? Why didn't they put the reference? The authors cannot put this statement here, without any scientific support! We should be careful with the strong statements we make, without any supporting scientific reference!
[line 83-87] The authors in the article state: “In Spain official doctoral studies are regulated by Royal Decree 99/2011 of January 28. A doctoral thesis is an original research work related with one of the fields of knowledge (Scientific, Humanist, Biomedical, Social and Technical), that the doctoral student will defend in front of a panel in public, once they have completed their training in a doctoral program.” - Is Royal Decree 99/2011, which refers to specific legislation in Spain, the transposition of a European Directive or Standard into Spanish domestic law? If so, it would be very important for the authors to add this information to the article...
[line 92-106] In the text of the article the reference [24] is missing. This should be reviewed and corrected by the authors!
[line 223-224] The authors in the article state: “(Research Procedure, Classification of Scientific Disciplines following Borms and Research Approach).” - but why didn't the authors put the reference here? It would be very important to have this information, so that in the future the scientific community can consult this reference.
Figure 5 with poor resolution... The authors need to improve the resolution of this "figure"...
[line 446-458] The authors make several recommendations for the future regarding what should happen with doctoral theses in Spain, according to the authors' opinion after completing this research. Which is very good! However, unfortunately the authors do not leave recommendations for the scientific community, about what work can be developed by future researchers, to continue the research developed by the authors. This would be very good for the article and for the authors!
This article has many old "References", some years old, which may raise some doubts, regarding the relevance and actuality of the theme researched by the authors.
The authors should have been more careful in selecting more current scientific articles!
Author Response
Dear Reviewer
Thank you for reviewing our manuscript
I am sending you an attachment with the requested corrections.
Kind regards
Antonio Antúnez

Reviewer 2 Report
See attached file

Author Response

(The authors gave the same response as above.)

Round 2
Reviewer 1 Report
The authors answered correctly to all my recommendations, but unfortunately they did not analyze the first recommendation: "The abstract is well written, but it could be improved if the authors included some of the results obtained, to create curiosity about the research to future readers.".
I am waiting for this situation to be analysed by the authors.
Author Response
- Response 1. We are very grateful for your suggestion and apologise for not having done so previously. We have included the following paragraph for completeness:
- “The scientific disciplines which presented the highest number of theses were Sport Sciences (n=33) and Sport Pedagogy (n=19). The main results show that quantitative approaches are the most common (n=49), doctoral theses using descriptive studies based on systematic observation represent the majority (73%), the predominant type of data collection was the use of cross-sectional studies (70.8%) versus longitudinal studies (26.4%), and the most used sampling method was that of convenience (n=65).”
